# Variation of extreme drought and flood in North China revealed by documentary-based seasonal precipitation reconstruction for the past 300 years

Jingyun Zheng[1,2], Yingzhuo Yu[1,2], Xuezhen Zhang[1,2], Zhixin Hao[1,2]

[1]Key Laboratory of Land Surface Pattern and Simulation, Institute of Geographic Sciences and Natural Resources Research, Chinese Academy of Sciences, Beijing 100101, China
[2]University of Chinese Academy of Sciences, Beijing 100049, China

*Correspondence to*: Zhixin Hao (haozx@igsnrr.ac.cn)

**Abstract.** Using the 17-sites seasonal precipitation reconstructions from an unique historical archive, Yu-Xue-Fen-Cun, the decadal variations of extreme droughts and floods (i.e., the event with occurrence probability of less than 10% from 1951 to 2000) in North China were investigated, by considering both the probabilities of droughts/floods occurrence in each site and spatial coverage (i.e., percentage of sites). Then, the possible linkages of extreme droughts and floods with ENSO (i.e., El Niño and La Niña) episodes and large volcanic eruptions were discussed. The results show that there were 29 extreme droughts and 28 extreme floods in North China from 1736 to 2000. For most of these extreme drought (flood) events, precipitation decreased (increased) evidently at most of sites for four seasons, especially for summer and autumn. But in drought years of 1902 and 1981, precipitation only decreased in summer slightly, while it decreased evidently in other three seasons. Similarly, the precipitation anomalies for different seasons at different sites also existed in several extreme flood years, such as 1794, 1823, 1867, 1872 and 1961. Extreme droughts occurred more frequently (2 or more events) during the 1770s–1780s, 1870s, 1900s–1930s and 1980s–1990s, among which the most frequent (3 events) occurred in the 1900s and the 1920s. More frequent extreme floods occurred in the 1770s, 1790s, 1820s, 1880s, 1910s and 1950s–1960s, among which the most frequent (4 events) occurred in the 1790s and 1880s. For the total of extreme droughts and floods, they were more frequent in the 1770s, 1790s, 1870s–1880s, 1900s–1930s and 1960s, and the highest frequency (5 events) occurred in the 1790s. A higher probability of the extreme drought was found when El Niño occurred in the current year or the previous year. However, no significant connections were found between the occurrences of extreme floods and ENSO episodes, or the occurrences of extreme droughts/floods and large volcanic eruptions.

## 1 Introduction

Extreme climate events, such as drought and flood, can lead to high impacts on the natural environment and social system, such as water resources, agriculture, economic activity and human health and well-being. Based on the evidences from observed data since 1950, the IPCC (2012) special report concluded that several regions in the world, particularly southern

Europe and West Africa, have experienced more intense and longer droughts; however, in central North America and northwestern Australia, droughts have become either less frequent, less intense, or shorter (with medium confidence). Meanwhile, there have been statistically significant increases in the number of heavy precipitation events (e.g., 95th percentile) in more regions than there have been statistically significant decreases over the world. Furthermore, the strong

regional and subregional variations exist in both of extreme droughts and heavy precipitation events. For example, in China, it has been shown that the droughts appeared more frequently in Northeast China, North China and the eastern part of Northwest China during 1961–2013, with persistent, severe and widespread droughts from the late 1990s to early 2000s. Moreover, severe droughts also became more and more frequent in Southwest China in 2006–2013. However, in the lower reaches of Yangtze River and the northern Xinjiang, the drought frequency tended to decrease from 1961 to 2013.

Meanwhile, in North China, the southwest part of Northeast China and the western Sichuan Basin, a downward trend occurred in yearly rainstorm (i.e., ≥50mm/d) days. In most of central and East China, rainstorm days showed an increasing trend (Qin et al., 2015).

However, the instrumental measurements generally covered more than a half century, which cannot represent the full natural climate variability in many regions of the world as those derived from paleo-climate reconstructions, especially for the

drought and flood (e.g., Cook et al., 2010; Ge et al., 2016; IPCC, 2012, 2013). Therefore, investigating variations in extreme climate events from long-term datasets is critical to identify whether the recent extreme events observed by instruments exceed the natural variability, which could provide more experience for adaptation to extremes and disasters in future (Qin et al., 2015), especially in regions with large precipitation variability and dense population, such as North China Plain (NCP). This region is located at the margin of the Eastern Asian summer monsoon (EASM) and has a sub-humid warm temperate

climate with the summer and autumn precipitation accounts for 80% annual precipitation approximately. As revealed by other studies (e.g., Wang, 2002; Wang and He, 2015; Ding and Wang, 2016), the climate in this region is sensitive to the global change and the rainfall decreased dramatically from the late 1970s, which had caused the evident impacts on water resources and agriculture when the EASM became weakened.

Recently, there were several studies focused on the historical severe drought/flood events in NCP. For example, based on "A

Compendium of Chinese Meteorological Records of the Last 3000 Years" (Zhang, 2004), Zhang (2005) identified the 15 severe persistent (≥ 3 years) drought events that occurred in NCP and surrounding area over the past 1000 years, and found that the most of historical persistent drought events (i.e., those before 1950) were more severe than that occurred in 1951–2000. Shen et al. (2008) investigated the characteristics of anomalous precipitation events during the past five centuries over eastern China (including North China and the mid-lower Yangtze River Valley), using the dataset of yearly dryness/wetness

grade over eastern China (CAMS, 1981), and found that in NCP, the high frequency of severe and extreme (5% and 2.5% occurrence probabilities in 1470–2000 respectively, same as for droughts) floods occurred in the 1650s, 1660s, 1750s, 1760s, 1820s and 1890s, whereas the period of 1580–1650 and the 1990s witnessed more severe and extreme droughts. Our previous studies (Hao et al., 2010a; Zheng et al., 2006) identified the extreme drought/flood events (10% occurrence probabilities in 1951–2000) and extreme persistent drought/flood events (≥ 3 years, 5% occurrence probabilities over all

2000 years) for the past 2000 years by merging the yearly dryness/wetness grade from individual sites. We found that there were more frequent extreme drought/flood (single year or persistent) events during the periods of 100–150, 550–650, 1050–1100, and 1850–1900 in NCP, while the frequency and intensity of extreme drought/flood events in the second half of the 20th century more closely resembled the mean status over the past 2000 years. Moreover, several other studies had focused

on individual events reconstructions with large impacts on agriculture and society, such as the persistent extreme droughts in 1784–1786 (Zhang, 2000), 1876–1878 (Hao et al., 2010b; Man, 2000; Zhang and Liang, 2010) and 1927–1930 (Zeng et al., 2009), and the extreme floods in 1730 (Zhang and Liang, 2016), 1755 (Zhang, 2012) and 1917 (Ma et al., 2015). Some earlier studies (e.g., Jiang et al., 1997; Qian et al., 2003a, b; Jiang et al., 2005) had also investigated the dry-wet variations for decadal to centennial-scale over East China for the last millennium using documentary-based reconstructions.

Meanwhile, several studies had argued that the anomalous precipitation in NCP, especially the occurrence of drought was related to El Niño and large volcanic eruption. For instance, from the observation data since 1951, it was found that the rainfall decreased over northern China (including NCP and the adjacent area at north and west) not only in the summer and autumn of El Niño developing year (Wu et al, 2003; Lu, 2005; Zhai et al., 2016), but also in the summer when El Niño decayed (Feng et al., 2014; Zhang et al., 2017). From the documentary-based reconstruction of yearly dryness/wetness grade

for the past 500 years, Chen and Yang (2013) argued that the occurrence of drought in northern China was in synchronous with El Niño events in the context of decadal variations. Shen et al. (2007) found that the most three exceptional drought events over eastern China occurred in 1586–1589, 1638–1641 and 1965–1966, with 50% or more summer rainfall reduction in the droughty centres, which might be triggered by large volcanic eruptions and amplified by the El Niño events.

However, most of these studies were performed from the yearly dryness/wetness grade data and relevant historical

descriptions or the limited instrumental period. Therefore, we present a case study, using seasonal precipitation reconstructions to investigate the variations of extreme drought and flood in NCP for the past 300 years, which is helpful to understand the impacts of seasonal-scale extreme climate on agriculture and social activities.

## 2 Data and method

### 2.1 Data

Three datasets were used in this study, including seasonal precipitation reconstruction, chronology of El Niño and La Niña events, and chronology of large volcanic eruption.

(1) Seasonal precipitation reconstruction. It included spring, summer, autumn and winter precipitation at 17 sites (Fig. 1) located in North China Plain (34°N–39°N, 108°E–120°E approximately) during 1736–2000 with annual resolution (Zheng et al., 2005). While 8 (i.e., Xi'an, Taiyuan, Shijiazhuang, Cangzhou, Ji'nan, Anyang, Weifang, and Linyi) of 17 sites had

complete records for the entire period (1736–2000), the other 9 sites involve some missing records in 1911–1950. This dataset was reconstructed from a unique historical archive called Yu-Xue-Fen-Cun, i.e., the quantitative records (in Chinese length units of "Fen" and "Cun"; 1 Fen = 0.32 cm approximately; 1 Cun = 10 Fens) of rainfall (i.e. "Yu") and snowfall (i.e.

"Xue") reported in memos to the emperor from local officials in Qing Dynasty (1644–1911), together with available instrumental data. Noting that the rainfall reported in Yue-Xue-Fen-Cun was measured as the infiltration depth (in units of Cun and Fen) from the dry-wet soil boundary layer to the ground surface by digging the soil with a shovel in the flat farmland after each rainfall event. The snowfall was measured as the depth on the surface after each snowfall event, which is
similar to the observation of snowfall depth at modern weather station. Besides the quantitative measurements of rainfall and snowfall, the Yu-Xue-Fen-Cun also reported other information, such as the dates and intensity for each precipitation event, the amount of rain or snow days, the summation of infiltration depth or snowfall depth, and the qualitative descriptions within a limited duration (e.g., one month or season) (Ge et al., 2005). For the rainfall reconstruction from historical records, the method was the Green-Ampt infiltration model under surface water balance equation:

$$P = (\theta_s - \theta_i) \times \rho \times Z_f / \beta$$

where $P$ is the precipitation; $\theta_s$, $\theta_i$, $\rho$, $Z_f$ and $\beta$ are the soil saturated moisture content, the initial moisture content before rainfall, the apparent specific gravity of the soil, the depth of infiltration (i.e. Yu-Fen-Cun in historical times), and the infiltration rate, respectively, which can be obtained from modern local agrometeorological stations based on the types of soil texture and the seasonality of climate at each site. This is because the physical properties of farmland soil and the
seasonality of climate are supposed to remain constant generally over the past 300 years. Moreover, the field infiltration experiment was used for the validation of reconstruction model, which showed that the predicted $R^2$ (i.e., explained variance) for the rainfall reconstruction reached to 87%. The snowfall reconstruction was calibrated by the regression equations based on the data of instrumental precipitation and snowfall depth from each weather station, which resulted in the predicted $R^2$ of 62–82% for different sites (Zheng et al., 2005). Such high predicted $R^2$ value enabled these reconstructions to adequately
capture a majority of the precipitation variability and extreme events.

(2) Chronology of El Niño and La Niña events. This chronology was reconstructed from tree-ring, ice-core, coral records and historical documents by Gergis and Fowler (2009). There were 119 El Niño and 127 La Niña events identified during 1736–2000. The magnitude of these El Niño/La Niña events was categorized into five grades with extreme (E), very strong (VS), strong (S), moderate (M), and weak (W). There are some other ENSO index reconstructions in the past millennium
(e.g., Stahle et al. 1998; Braganza et al. 2009; McGregor et al. 2010; Wilson et al. 2010; Li et al. 2011, 2013). The reason for our study using the reconstruction by Gergis and Fowler (2009) is that their result was compiled as a chronology for each El Niño and La Niña episode rather than the ENSO index; thus, it is more appropriate for comparison with the extreme drought/flood event by event.

(3) Chronology of large volcanic eruption. The chronology of large volcanic eruption used in this study was extracted from
the database of "Volcanoes of the World" released by the Smithsonian Institution (Global Volcanism Program, 2013). This dataset includes information on volcano location (longitude, latitude and elevation), starting and ending dates of eruptive activity, tephra volume, and volcanic explosivity index (VEI) for each eruption, in which the VEI is determined by the eruption type and duration, the tephra volume, and the height of the eruption cloud column. Compared to the other reconstruction on volcanic eruption index (e.g., Sigl et al., 2015), this chronology contains each volcanic eruption event,

which is convenient to compare with the extreme drought/flood event. Noted that only the eruptions with VEI ≥ 4 were extracted as large eruptions for this study, and there were 137 large eruptions occurred in 1736–2000.

## 2.2 Method

Firstly, we calculate the threshold for probability of 10%, 20%, 80% and 90% occurrence based on the 17-site precipitation
reconstruction series according to Gamma distribution, to identify the year when the severe or extreme drought/flood occurred in period of 1736–2000. For each site, the severe drought (or flood) means that the annual precipitation was lower (or higher) than the threshold for probability of 20% (or 80%), and the extreme drought (or flood) was defined as below (or above) the threshold for probability of 10% (or 90%). Then, we calculate the percentage of sites with extreme and severe drought (or flood) occurred in the study area (Fig. 2). It's shown that the top five (i.e., 10% occurrence) drought events
between 1951–2000 (i.e., instrumental period) occurred in 1997, 1986, 1965, 1981 and 1991; and the top five flood years were 1964, 1958, 1963, 1956 and 1961 (Fig. 2a). Therefore, we use the minimum percentage of severe and extreme drought (flood) sites among these extreme years in 1951–2000, i.e., 35% (35%) and 29% (24%) of all sites experiencing severe and extreme drought (flood) respectively, as the criteria to identify the regional extreme drought/flood events in 1736–2000 (Fig. 2b).
Furthermore, we compare the extreme drought/flood events with the El Niño/La Niña chronology and the large volcanic eruption chronology using the contingency table, to illustrate the characteristics of connections between extreme drought/flood events, and El Niño/La Niña episodes, and large volcanic eruptions, respectively. For example, to detect whether the frequency of extreme drought becomes higher in the years after El Niño events, we create the contingency table by calculating the numbers of occurrences with extreme drought and El Niño in the previous year, extreme drought and no
El Niño in the previous year, no extreme drought but El Niño in the previous year, no extreme drought and El Niño in the previous year. Then, the Chi-square test ($\chi^2$) is adopted to test the significance. Similarly, the contingency table and test are also performed for other cases with the occurrences of extreme drought or flood events in NCP associated with the events of ENSO or large volcanic eruption, respectively.

## 3 Results and discussion

### 3.1 Occurrence of extreme drought and flood

There were 29 extreme drought events and 28 extreme flood events identified (Fig. 2) in 1736–2000. Extreme drought events occurred in 1743, 1777, 1778, 1783, 1786, 1792, 1805, 1813, 1847, 1856, 1869, 1876, 1877, 1900, 1901, 1902, 1916, 1919, 1920, 1922, 1927, 1936, 1939, 1941, 1965, 1981, 1986, 1991 and 1997. Extreme flood events occurred in 1742, 1751, 1774, 1776, 1794, 1797, 1798, 1799, 1800, 1822, 1823, 1830, 1858, 1867, 1872, 1882, 1883, 1886, 1889, 1890, 1910, 1914, 1937,
1956, 1958, 1961, 1963 and 1964. Figure 3 illustrates the box-whisker plot of seasonal precipitation anomaly percentage among 17 sites for each extreme drought and flood event. It's shown that for a majority of extreme drought (flood) events,

precipitation decreased (increased) evidently at most of sites for four seasons, especially for summer and autumn, because the precipitation in summer and autumn accounts for approximately 60% and 20% of the annual precipitation, respectively. For example, at the extreme drought year of 1877, the regional precipitation anomaly (i.e., referenced to the average of all sites over entire study area relative to the mean precipitation of all years) was -25% in spring, -53% in summer, -53% in autumn, and -23% in winter. In the extreme flood year of 1890, the regional precipitation anomaly was 37% in spring, 32% in summer, 23% in autumn, and 30% in winter. Nevertheless, in drought years of 1902 and 1981, precipitation only decreased in summer slightly, while it decreased evidently in other three seasons. Similarly, the precipitation anomalies for different seasons at different sites also existed in several extreme flood years, such as 1794, 1823, 1867, 1872 and 1961.

Compared to the extreme droughts and floods reported in previous studies (Chen and Yang, 2013; Hao et al., 2010a, b; Shen et al., 2007, 2008; Zhang, 2005; Zheng et al., 2006), our results identified a majority of extreme drought years and 10 extreme flood years (i.e., 1751, 1800, 1822, 1823, 1883, 1889, 1937, 1956, 1963 and 1964) in their publications. Moreover, our results revealed 9 extreme drought events and 18 extreme flood events (marked "↑" in Fig. 2) which were not reported before. Meanwhile, our results also eliminated those events that the intensity may have been overestimated by the previous studies, which resulted from droughts/floods only at sub-regional scale or for a short duration. This is mainly because the series of seasonal precipitation were reconstructed from Yu-Xue-Fen-Cun, thus more accurate in seasons and sub-regions than the series of yearly dryness/wetness grade used in other studies. For example, 1785 and 1825 were reported as extreme drought years by Chen and Yang (2013) and Hao et al. (2010a), respectively. However, the drought in 1785 occurred only from late spring to early summer at several sites in the southern part of NCP, and it did not prevail over the entire study area. The drought of 1825 occurred in summer over approximately half of NCP, but in spring, rainfall increased by more than 50% across nearly the entire region. Similar situations also occurred in 1826, 1832, 1846, 1878, 1899, 1928, 1929 and 1972. Meanwhile, 1757, 1761, 1819, 1894, 1898 and 1973 were reported as extreme flood years by Shen et al. (2008) and Hao et al. (2010a). However, in 1757, precipitation only increased significantly in the southern of NCP, except that more snowfall occurred at most sites of the entire study area in winter. In 1761, floods occurred over almost half of NCP in spring only; but only 2 sites occurred severe drought and 3 sites experienced extreme flood throughout the whole year. In 1819 and 1894, although more snowfall occurred for most sites in winter, there were only a few sites experiencing severe flood, and no sites had extreme flood throughout the year. In 1898, more precipitation only occurred in spring but not for other seasons. In 1973, only 6 of 17 sites experienced severe floods but no extreme floods occurred at any sites throughout the year, despite 1 site of them had extreme rainfall during summer time. Thus, these years cannot be identified as extreme flood events in our result.

### 3.2 Variation of extreme droughts and floods

Figure 4 illustrated the frequency of extreme drought and flood in NCP for each decade from 1740s to 1990s. It showed that the extreme drought occurred more frequently in the 1770s–1780s, 1870s, 1900s–1930s and 1980s–1990s with at least 2 extreme events for each decade, and the two decades of the 1900s and the 1920s both had 3 events. Moreover, some of them occurred within 2–3 years on a roll, e.g. in 1777–1778, 1876–1877, 1900–1902, 1919–1920. These consecutive events

usually caused severe impacts on agriculture and society. For example, the droughts in 1876–1877 led to evidently poor harvests with reduction of about 45% and 50% in 1876 and 1877 respectively (Hao et al., 2010b). Even worse, this consecutive extreme drought further caused evidently delayed sowing and crop failure within several years after 1877, and led to the rice price increased by 5–10 times than that in the normal year (Hao et al, 2010b). Such persistent and large spatial poor harvests and food scarcities not only caused more than one hundred million people in famine, but also triggered more than one hundred thousand refugees to emigrate from NCP to eastern Inner Mongolia (Xiao et al., 2011b), which finally resulted in more than 13 million people died from famine and plague (Li et al., 1994). However, there were no extreme droughts in the 1750s–1760s, 1820s–1830s, 1880s–1890s, 1950s and 1970s. Besides, the frequency of extreme drought showed a slightly increase trend (0.29 times per 100-yr) from 1740s to 1990s, but it was not statistically significant.

More frequent extreme floods occurred in the 1770s, 1790s, 1820s, 1880s, 1910s and 1950s–1960s with two or more occurrences per decade, but no extreme flood occurred in the 1760s, 1780s, 1810s, 1840s, 1900s, 1920s, 1940s and 1970s–1990s. Meanwhile, the most frequent extreme floods (4 events) occurred in the 1790s and the 1880s respectively, in which the consecutive extreme flood years in 1797–1800 caused flowages from several rivers and resulted that approximately 1/4 of the total counties in North China Plain were flooded (Zheng et al., 2016). Moreover, 4 extreme flood years in the 1880s caused the continuous breaching of the dyke along the Yellow River from 1882 to 1890 (Zhang, 2010).

For the extreme drought and flood events in total, most of them occurred in the 1770s and 1790s, 1870s–1880s, 1900s–1930s and 1960s, among which the 1790s witnessed the highest frequency of extreme drought and flood events. Such frequently extreme droughts and floods, together with climate cooling from the late of 18th century, resulted that the regional socioeconomic system became more vulnerable around the turn of the 19th century in NCP (Fang et al, 2013). Furthermore, more frequent extreme floods/droughts caused many negative impacts, e.g., vulnerable food security, significant increase of disaster victims, which led to the deterioration of refugee relief and more occurrence of peasant uprising (Xiao et al., 2011a). However, no extreme drought or flood event occurred in the 1760s and the 1970s.

### 3.3 Probabilities of the occurrences of extreme events with ENSO events and large volcanic eruptions

Table 1 shows the occurrences of ENSO (i.e., El Niño and La Niña events) and large volcanic eruptions (i.e., VEI ≥ 4) in the extreme drought/flood years and their previous years. It's shown that among the years before 29 extreme droughts, 19 of them were El Niño years, 8 of them were La Niña years, 2 of them did not experience ENSO event; and 10 of them experienced large volcanic eruptions. Among the 29 extreme drought years, 17 occurred in El Niño years, 8 occurred in La Niña years, 4 years did not experience ENSO events; and 12 years experienced large volcanic eruptions (Table 2). Among the years before 28 extreme flood events, 11 of them were El Niño years, 9 of them were La Niña years, 8 of them did not experience ENSO events; and 11 of them experienced large volcanic eruptions. Among the 28 extreme flood years, there were 8 El Niño years, 13 La Niña years, 7 years did not experience ENSO events; and 12 years experienced large volcanic eruptions (Table 2). In total, 17 of the 29 extreme drought events and 18 of the 28 extreme flood events coincided with the occurrence of large volcanic eruptions in either the same or the previous year (Table 2).

The Chi-square test ($\chi^2$) showed that there is a higher probability of the extreme drought with the El Niño occurrence in the same year or the previous year. For example, the chi-square value is 7.997 for the occurrence of extreme drought and El Niño in the previous year, which is significant at the $p<0.01$ level. Regarding the occurrence of extreme drought and El Niño in the same year, the Chi-square value is 4.502, which passes the $p<0.05$ significant level. Hao et al (2008, 2010b) found that the precipitation over the NCP in the El Niño year or the sequent year was below that in normal years, and the severe drought of 1876–1877 was associated with the strong El Niño episode. Chen and Yang (2013) and Li et al. (2011) also found that most of drought events or extreme dry years in northern China might have a close link with the occurrence of El Niño during historical times. In addition, many previous studies from observation found that the El Niño could usually cause precipitation evidently decreasing in northern China not only simultaneously, but also in the subsequent summer after an El Niño year (e.g., Wu et al., 2003; Lu, 2005, Zhang et al, 2017). Corresponding to the recent strong El Niño event in 2015-2016, the summer precipitation decreased by 20% to 50% over the northern China (Zhai et al. 2016). Our result confirmed their findings. As suggested by previous studies based on observations and simulations, the mechanism of impact of El Niño on precipitation in NCP can be summarized as follows. In the developing stage of El Niño episode, weakened Walker Circulation could restrain the India summer monsoon and further trigger up an anomalous barotropic cyclone over the East Asian by modulating the wind fields from the western of Tibetan Plateau and affecting the mid-latitude Asian wave pattern along 30°N–50°N. Correspondingly, NCP is located at the area affected by the local downdraft airflow of that anomalous barotropic cyclone. Thus, the rainfall decreases at summer and autumn over NCP (Wu et al, 2003; Lu, 2005). While in the peak and decaying year of El Niño, the warm SST anomaly in tropical Indian Ocean can trigger up Kelvin waves to induce an anomalous easterlies in equatorial atmosphere and the anomalous anticyclone over western North Pacific (Xie et al., 2009). Since the southern flank of western Pacific subtropical high (WPSH) prevail easterly winds, the enhanced easterlies lead to the southward shift of WPSH (Song and Zhou, 2014). The southward shift of WPSH consequently dampens the moisture transport from the Indian summer monsoon to eastern China, resulting in a significant decrease of rainfall in NCP (Zhang et al., 2017).

However, the Chi-square test demonstrated that there is no significant connection for the occurrences between extreme flood and ENSO events (Table 2). In addition, the Chi-square test also suggested that no significant link exists between the occurrences of extreme drought/flood events and large volcanic eruptions, although Shen et al. (2007) had argued that the large volcanic eruptions might trigger the exceptional drought events over eastern China.

## 4 Conclusions

This study investigated the decadal variation of extreme drought/flood over North China based on the 17-sites seasonal precipitation reconstructions in 1736–2000, in which the extreme drought/flood events were defined as those with occurrence probability lower than 10% in reference period of 1951–2000, by considering the probability of drought/flood occurrence in each site and spatial coverage together. It's found that there were 29 extreme droughts and 28 extreme floods

in North China during 1736–2000, in which precipitation decreased (increased) evidently in most sites for all seasons, especially in summer and autumn for most of them. In 1777–1778, 1876–1877, 1900–1902 and 1919–1920, the extreme droughts occurred sequentially, and in 1797–1800, 1882–1883, 1889–1890 and 1963–1964, the extreme floods appeared in a roll. Compared to the previous studies on extreme droughts and floods derived from the yearly dryness/wetness grade data,

this study found 9 extreme drought events and 18 extreme flood events that had not been reported previously.

Moreover, the results showed that there was an evidently decadal variation in the occurrence of extreme drought/flood events from 1740s to 1990s. In the 1770s–1780s, 1870s, 1900s–1930s and 1980s–1990s, the extreme drought occurred at least 2 times in each decade, among which the most frequent occurrences (3 times) were in the 1900s and the 1920s. While twice or more extreme floods occurred in the 1770s, 1790s, 1820s, 1880s, 1910s and 1950s–1960s, with 1790s and 1880s having the

most frequent occurrences (4 times). As the extreme drought and flood events in total, there were more frequent in the 1770s, 1790s, 1870s–1880s, 1900s–1930s and the 1960s, and the most frequent (5 events) decade were in the 1790s. In addition, comparison on the occurrences of extreme drought/flood with the chronologies of ENSO and large volcanic eruption by the Chi-square test ($\chi^2$) confirmed that there was a higher probability of the extreme drought followed the El Niño episode. Nevertheless, no significant connection existed either between the occurrences of extreme flood and El Niño/La Niña

episodes or between the occurrences of extreme drought/flood and large volcanic eruption.

The archive of Yu-Xue-Fen-Cun provided the quantitative records on rainfall and snowfall with high spatial and temporal resolution for seasonal precipitation reconstruction from 1736, which enabled us to investigate the variation of extreme drought and flood with seasonal features. Besides Yu-Xue-Fen-Cun, China also has other historical documents (e.g., local gazettes, official histories, weather diaries, etc.) with abundant and well-dated records on weather, anomalous climate,

climate-related natural disasters, the impacts of weather and climate anomalies, as well as phenology, which have been used for the reconstruction of past climates extended to more than thousands of years at resolution of annual to decadal scale (Ge et al., 2016). Thus, most of them could further be applied to identify the regional extreme climate events at yearly resolution or prominent decadal climate anomalies before 1736, and to investigate long-term pattern on the occurrences of regional extreme climate events associated with anomalous forcings in future works.


**Competing interests.** The authors declare that they have no conflict of interest.

**Acknowledgements.** This study was supported by the National Key R&D Program of China (2016YFA0600702), the National Natural Science Foundation of China (No.41430528, 41671201), and the Chinese Academy of Sciences

(XDA19040101).

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

**Figures**

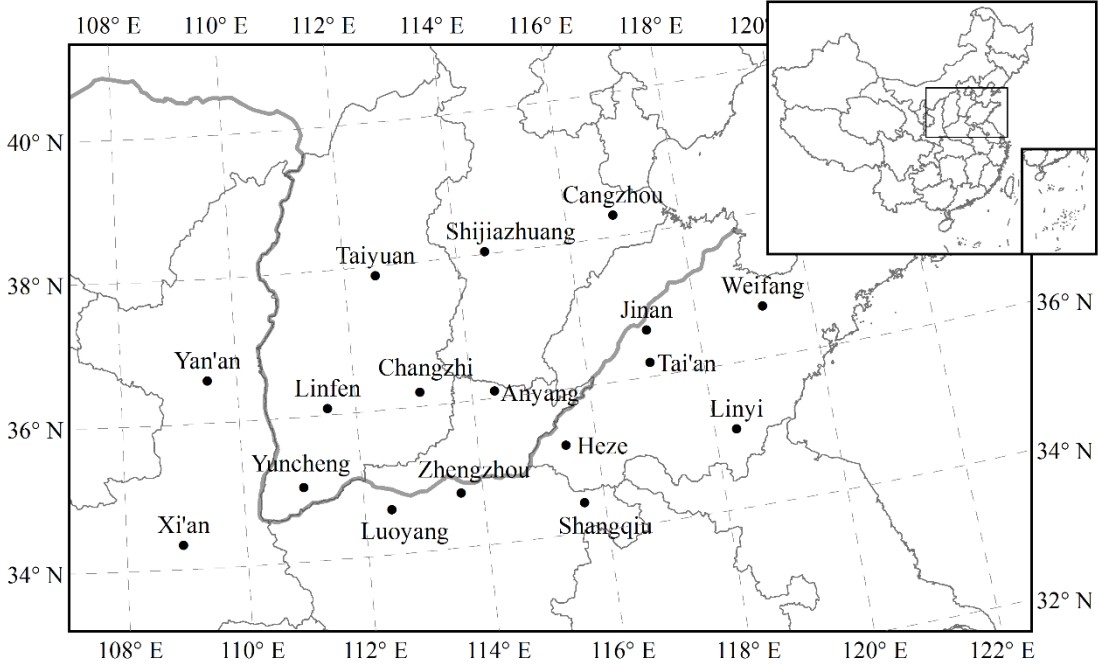

**Figure 1. The location of 17 sites with seasonal precipitation reconstruction for 1736–2000.**

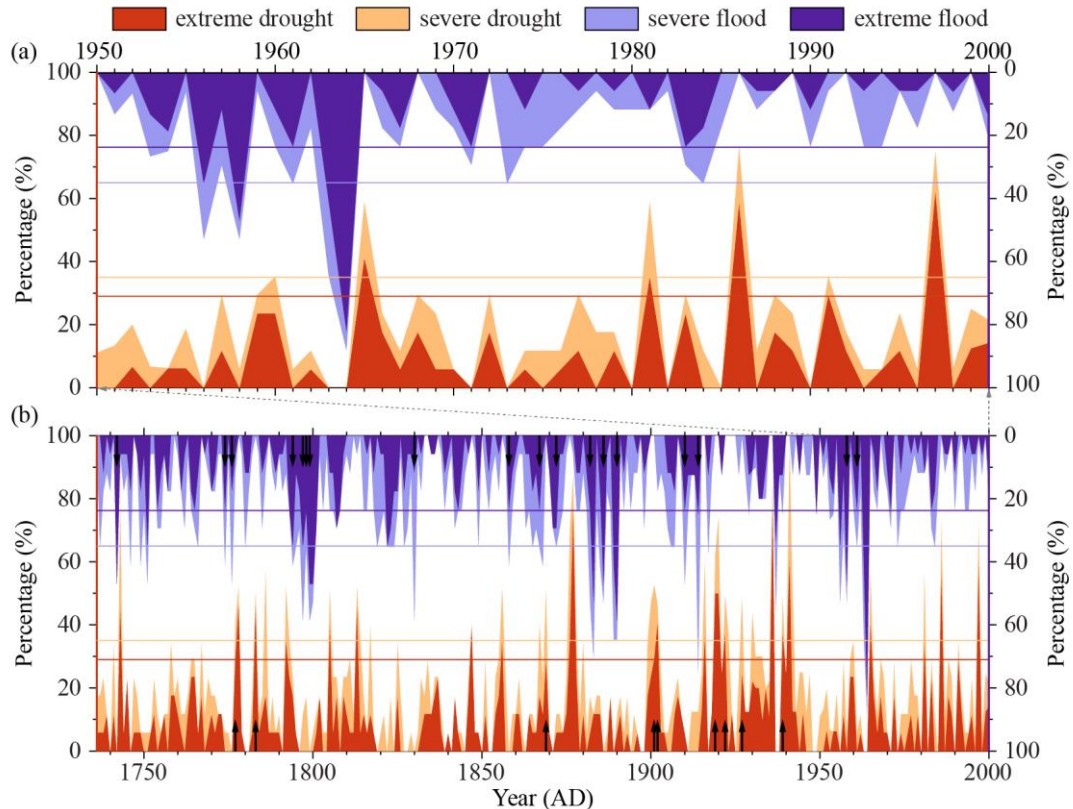

**Figure 2. Percentage of sites with extreme and severe drought/flood occurred over North China in 1736–2000. Dash line: the criteria to identify the regional extreme drought/flood events respectively. ↑: The year of extreme drought/flood events which were not reported in previous studies (Chen and Yang, 2013; Hao et al., 2010a, b; Shen et al., 2007, 2008; Zhang, 2005; Zheng et al., 2006).**

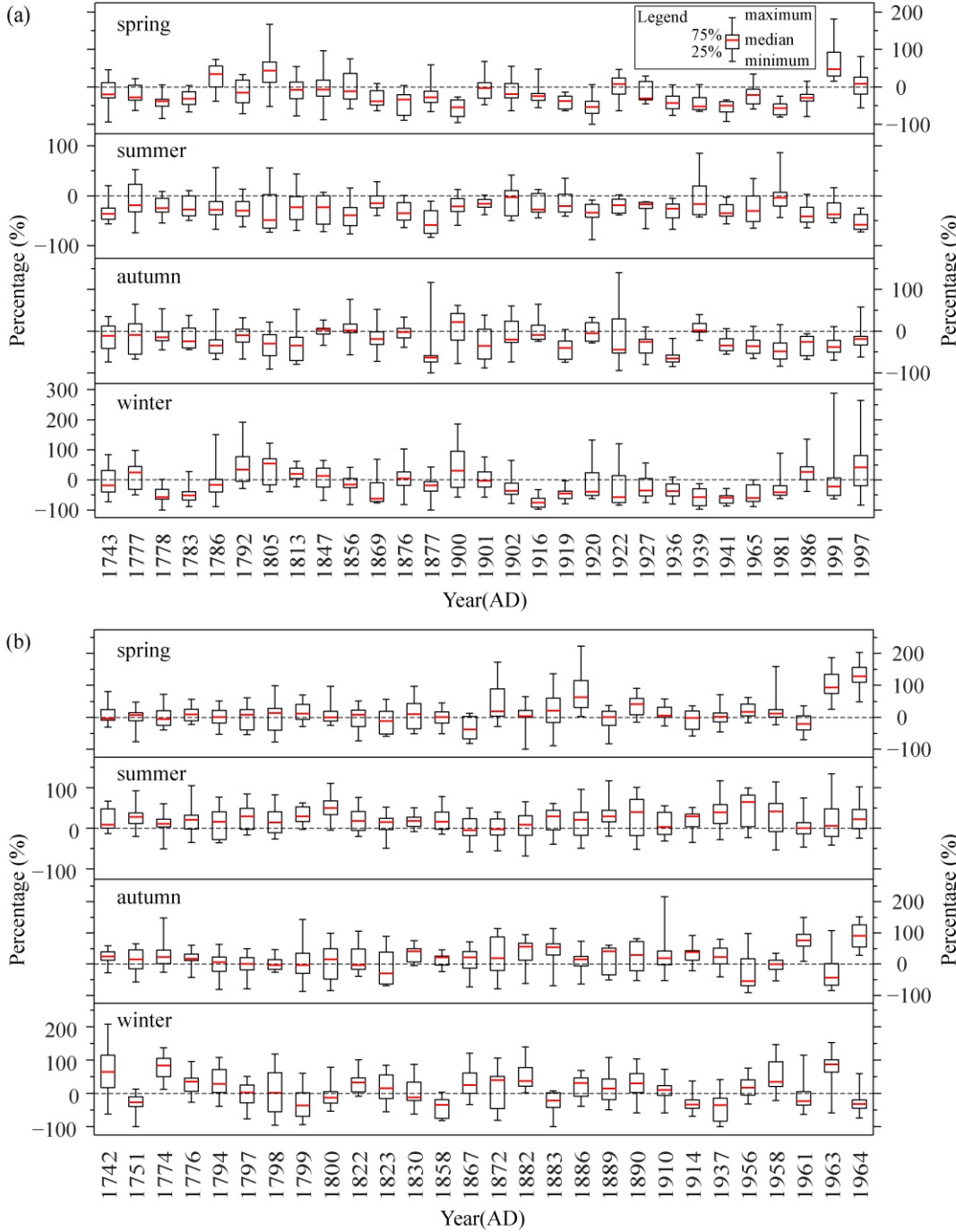

**Figure 3.** The box-whisker plot of seasonal precipitation anomaly percentage among sites for each extreme drought (a) and flood (b) event.

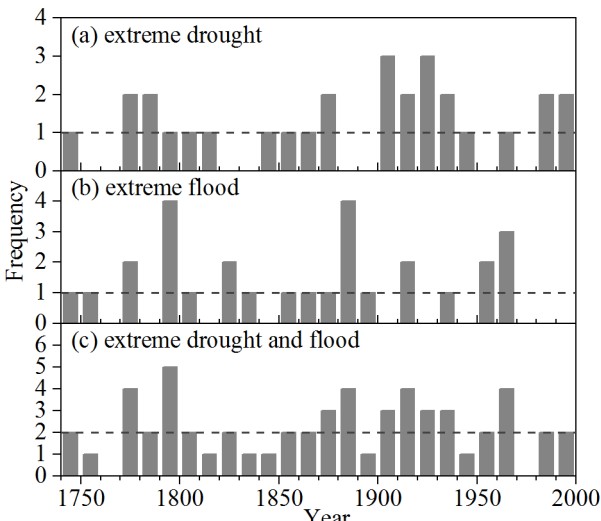

**Figure 4. The frequency of extreme drought and flood in North China for each decades from 1740s to 1990s.**

**Tables**

**Table 1. The occurrences of El Niño and La Niña event and large volcanic eruption in the extreme drought/flood year and its previous year**

| Extreme drought year | El Niño/La Niña occurred | Large volcanic eruptions (year/month) | Extreme flood year | El Niño/La Niña occurred | Large volcanic eruptions (year/month) |
|---|---|---|---|---|---|
| 1743 | $L_E$/1742→$L_{VS}$/1743 | - | 1742 | $L_S$/1741→$L_E$/1742 | 1741/8 |
| 1777 | $L_W$/1776→$E_W$/1777 | - | 1751 | $L_S$/1750→$L_M$/1751 | 1750/? |
| 1778 | $E_W$/1777→$L_W$/1778 | 1778/? | 1774 | $L_W$/1773 | - |
| 1783 | $E_W$/1782→$E_M$/1783 | 1783/6, 1783/8 | 1776 | $L_W$/1776 | - |
| 1786 | $L_W$/1785→$L_S$/1786 | 1786/? | 1794 | $E_M$/1793→$E_M$/1794 | 1793/2, 1793/3 |
| 1792 | $E_{VS}$/1791→$E_W$/1792 | - | 1797 | $L_M$/1797 | - |
| 1805 | $E_W$/1804→$L_{VS}$/1805 | - | 1798 | $L_M$/1797→$L_W$/1798 | - |
| 1813 | $E_S$/1812→$L_M$/1813 | 1812/4, 1812/8, 1813/? | 1799 | $E_W$/1798→$E_S$/1799 | - |
| 1847 | $E_W$/1846→$L_S$/1847 | 1846/6 | 1800 | $E_S$/1799 | 1800/1, 1800/? |
| 1856 | $E_M$/1856 | 1856/9 | 1822 | - | 1822/3, 1822/10 |
| 1869 | $E_{VS}$/1868 | - | 1823 | $L_M$/1823 | 1822/3, 1822/10 |
| 1876 | $L_S$/1875→$E_W$/1876 | 1875/3 | 1830 | $E_W$/1829 | 1829/9 |
| 1877 | $E_W$/1876→$E_{VS}$/1877 | 1877/6, 1877/? | 1858 | $L_W$/1857→$E_M$/1858 | 1857/1 |
| 1900 | $E_S$/1899→$E_{VS}$/1900 | 1899/11 | 1867 | $L_M$/1866→$L_S$/1867 | - |
| 1901 | $E_{VS}$/1900→$E_S$/1901 | - | 1872 | $L_{VS}$/1871→$L_M$/1872 | 1872/4, 1872/? |
| 1902 | $E_S$/1901→$E_{VS}$/1902 | 1901/5, 1901/5, 1901/5, 1901/10 | 1882 | $E_M$/1881 | - |
| 1916 | $E_{VS}$/1915→$L_S$/1916 | - | 1883 | - | 1883/8, 1883/10 |
| 1919 | $E_{VS}$/1918→$E_S$/1919 | 1918/4, 1918/10, 1919/5, 1919/8 | 1886 | $E_S$/1885→$L_M$/1886 | 1886/1, 1886/6, 1886/8 |
| 1920 | $E_S$/1919→$E_W$/1920 | 1919/5, 1919/8 | 1889 | $E_{VS}$/1888→$E_W$/1889 | 1888/7, 1889/10 |
| 1922 | $L_W$/1921→$L_S$/1922 | - | 1890 | $E_W$/1889→$L_S$/1890 | 1889/10, 1890/2 |
| 1927 | $E_E$/1926 | 1926/4 | 1910 | $L_{VS}$/1909→$L_{VS}$/1910 | - |
| 1936 | $E_W$/1935 | - | 1914 | $E_{VS}$/1913→$E_{VS}$/1914 | 1913/1, 1914/1 |
| 1939 | $E_M$/1938→$E_M$/1939 | - | 1937 | $E_W$/1937 | 1937/5 |
| 1941 | $E_{VS}$/1940→$E_E$/1941 | - | 1956 | $L_S$/1955→$L_M$/1956 | 1955/7, 1956/3 |
| 1965 | $L_W$/1964→$E_S$/1965 | 1964/11, 1965/9 | 1958 | $E_S$/1957→$E_S$/1958 | - |
| 1981 | $E_W$/1980 | 1980/5, 1981/4, 1981/5 | 1961 | - | - |
| 1986 | $L_M$/1985→$E_M$/1986 | 1986/3, 1986/11 | 1963 | $E_M$/1963 | 1963/3, 1963/5 |
| 1991 | $E_S$/1991 | 1990/1, 1990/2, 1991/6, 1991/8 | 1964 | $E_M$/1963→*$L_W$/1964 | 1963/3, 1963/5, 1964/11 |
| 1997 | $L_W$/1996→$E_{VS}$/1997 | - | | | |

5   E represents El Niño, L represents La Niña, and the subscripts represent their magnitudes, which were categorized into five grades: extreme (E), very strong (VS), strong (S), moderate (M), and weak (W).The symbol '-' means that neither an El Niño/La Niña event nor a large volcanic eruption occurred.

* In 1964, El Niño lasted until March, then changed into a La Niña in April.

**Table 2. The probabilities of extreme events occurrences with ENSO events and large volcanic eruptions**

| Occurrence | | Extreme drought | Extreme flood | No extreme drought/flood |
|---|---|---|---|---|
| | Total number | 29 | 28 | 208 |
| ENSO in the previous year | El Niño | 19 (65.5%)*** | 11 (39.3%) | 79 (38.0%) |
| | La Niña | 8 (27.6%) | 9 (32.1%) | 90 (43.3%) |
| | No El Niño/La Niña | 2 (6.9%) | 8 (28.6%) | 39 (18.8%) |
| ENSO in the very year | El Niño | 17 (58.6%)** | 8 (28.6%) | 82 (39.4%) |
| | La Niña | 8 (27.6%) | 13 (46.4%) | 90 (43.2%) |
| | No El Niño/La Niña | 4 (13.8%) | 7 (25.0%) | 36 (17.3%) |
| Large volcanic eruption in | the previous year | 10 (34.4%) | 11 (39.3%) | 85 (40.9%) |
| | the very year | 12 (41.4%) | 12 (42.9%) | 83 (39.9%) |
| | the very year or the previous year | 17 (58.6%) | 18 (64.3%) | 129 (62.0%) |
| | No eruptions | 12 (41.4%) | 10 (35.7%) | 79 (38.0%) |

The significant level of Chi-test ($\chi^2$), ***: p<0.01; **: p<0.05.