# Peer review of "Variation of extreme drought and flood in North China revealed by documentary-based seasonal precipitation reconstruction for the past 300 years"

_Climate of the Past, 2018_

## Short Comment (SC1) · 24 Apr 2018

Dear authors,

with interest I saw your manuscript on extreme droughts and floods in northern China over the past 300 years.

As this touches on a topic of interest of mine, I would like to mention a few older papers which may be of interest to your use of documentary evidence:

[Figure]

Jiang, J. , Zhang, D. and Fraedrich, K. (1997), Historic climate variability of wetness in East China (960–1992): a wavelet analysis. Int. J. Climatol., 17: 969-981. doi:10.1002/(SICI)1097-0088(199707)17:9<969::AID-JOC171>3.0.CO;2-B

Qian, W., Hu, Q., Zhu, Y. et al. (2003) Centennial-scale dry-wet variations in East Asia, Climate Dynamics, 21: 77. https://doi.org/10.1007/s00382-003-0319-3

Qian, WH., Chen, D., Zhu, Y. et al. (2003) Temporal and spatial variability of dryness/wetness in China during the last 530 years. Theor Appl Climatol, 76: 13. https://doi.org/10.1007/s00704-003-0009-4

Jiang, T., Zhang, Q., Blender, R. et al. (2005) Yangtze Delta floods and droughts of the last millennium: Abrupt changes and long term memory. Theor. Appl. Climatol., 82: 131. https://doi.org/10.1007/s00704-005-0125-4

Yours Sincerely
* * *

---

## Referee Comment (RC1) · Anonymous Referee #1 · 1 May 2018

General comments In this paper, extreme drought and flood events in North China during 1736-2000 have been identified with seasonal precipitation reconstructions of 17 sites, and then compared with ENSO episodes and large volcanic eruptions. Higher temporary and spatial resolution of precipitation reconstructions allows the authors to identify more extreme events than the previous studies and eliminate some ones that intensity may have been overestimated (as discussed in the section 3.1). This work is helpful for better understanding of the climatic extremes in the past and their driving mechanisms. Before acceptance and publication, some minor revisions are still

necessary.

Specific comments 1. This paper contains two main parts, identification of extreme drought/flood events, and the relationship between these events and ENSO/volcanic eruptions, however, in the Introduction (Section 1), the authors just introduce the study progress of historical climate change and extreme events. The previous studies cited in the Section 3.3 (from page 6 line 26 to page 7 line 2) which are about the relationship between climatic extremes and ENSO/volcanic eruptions could be moved into the Introduction, and more relevant literature should be added. 2. In the Section 2.1, the authors introduce the seasonal precipitation reconstruction which is from their previous work (page 3 line 10-23), and more details of this work, especially the description of the quantitative precipitation records of Yu-Xue-Fen-Cun during the Qing Dynasty and the methods of reconstruction, would be helpful for better evaluation of the scientific significance of this paper. 3. In the Section 2.2 (Method), the method how the authors identify the extreme drought/flood events should be described more clearly. For example, what does the phrase "these years" (in page 4 line 9-10) mean, all the years in 1736-2000, or just the 10 extreme years in 1951-2000 mentioned in the last sentence? The methods used for comparing the extreme drought/flood events with the ENSO chronology and the large volcanic eruption chronology also need some supplements. 4. In the Section 3.3, the authors give the probabilities of the occurrences of extreme events with ENSO events and large volcanic eruptions, however, no driving mechanism. Further discussion on how ENSO and volcanic eruption impact the climate change and extreme events might be necessary for this paper. What are the opinions in the previous studies, and why the results of this paper are different?

Technical corrections 1. This paper might need to be read and corrected by a native English speaker. 2. In page 4 line 10, the sentence "35% (35%) and 29% (24%) for severe and extreme drought (flood) respectively" should be checked, because according to the Fig.2, 35% (35%) is probably the criteria to identify the extreme drought (flood) events, but not severe ones. 3. For the Fig.3 (page 13), a legend might be necessary

to describe the symbols appearing on the figure.

---

## Referee Comment (RC2) · Anonymous Referee #2 · 19 May 2018

Dear editor and authors of the manuscript "Decadal variation of extreme drought and flood in North China revealed by documentary-based seasonal precipitation reconstruction for the past 300 years". To the best of my knowledge, the Yu-Xue-Fen-Cun record is the best proxy data in Eastern China with seasonal resolution and significant relationship to the instrumental precipitation variability. This work chooses the study area at the margin of the Eastern Asian summer monsoon, which will have a critical implication to understand the monsoon variability over the past several hundred years. The idea to definite and identify the extreme drought/flood events during the

different centuries is novel in paleoclimate reconstruction community. The results are completely supported by the data analysis. Thus, I suggest that the manuscript should be accepted for publication after a minor revision.

Main comments:

1. The explanation of the features of the extreme drought/flood events in North China could be further inferred. e.g. 1) The higher probability of the extreme drought events mostly followed the El Niño activity, which implied that the El Niño' may triggered the drought event in North China. 2) The precipitation variability in summer and autumn is stronger than other seasons, the reason may be that the precipitation during the flood season accounts for a large proportion of annual precipitation. 3) Some events have not mentioned in the previous studies, which indicates that the Yu-Xue-Fen-Cun has more accurate than other proxy record, e.g. the drought/flood grads, or the definition of the drought/flood events in this study is different from the other studies.

2. The phrase 'decadal variation' usually means that the record is smoothed by the 7 or 9 years low-pass filter to show the decadal variability. Here, the inter-annual events are analyzed during the different centuries. Thus, the phrase 'decadal variation' may not be a best choice.

3. The reason for selecting the study area should be further emphasized in climate research, e.g. the North China is located at the margin of the Eastern Asian summer monsoon, which is more sensitive to the global climate change.

Specific Comments:

1. Page 1, The seasonal features of the drought/flood events should be added in the abstract to emphasize the advantage of the Yu-Xue-Fen-Cun record.

2. Page 2, line 14. Is the 'larger precipitation anomaly percentage' more suitable? The reason is that the precipitation in south China is larger than one in North China.

3. Page 2, lines 16-35. The definition of drought/flood events should be added since

the different definition will obtain the different classifications. e.g. Zhang (2005) points out that there are only 15 events in North China over the past 1000 years in Page 2, lines 18-19, but there are five events during some centuries in this study.

4. Page 3, lines 1-3. The Yu-Xue-Fen-Cun record is not firstly used in this study, thus, this sentence is inappropriate.

5. Page 3, line 13. Here, the longitude should be a range from 108°E to another longitude (120°E?).

6. Page 3, lines 23-24. Please explain the reason for selecting Gergis and Fowler's ENSO index, because there are some other ENSO index reconstructions in the past millennium (Braganza et al. 2009; Cook et al. 2008; Li et al. 2011; Li et al. 2013; McGregor et al. 2010; Stahle et al. 1998; Wilson et al. 2010).

7. Page 3, lines 27-28. Regarding to the volcanic eruption index, a new reconstruction is encouraged to use (Sigl et al. 2015). Moreover, it would be interesting that the other degrees of eruptions are assessed, or the Southern Hemisphere eruptions are excluded.

8. Page 5, line 11. There is an extra period '.'.

9. Please change 'El Nino' with 'El Niño'.

10. If the Yu-Xue-Fen-Cun record is archived in a published repository, which will be a huge contribution for the paleoclimate integration community.

References:

1. Braganza K, Gergis JL, Power SB, Risbey JS, Fowler AM (2009) A multiproxy index of the El Niño–Southern Oscillation, A.D. 1525–1982. J Geophys Res-Atmos 114. doi:10.1029/2008JD010896

2. Cook ER, D'Arrigo RD, Anchukaitis KJ, Diaz HF (2008) ENSO reconstructions from long tree-ring chronologies: Unifying the differences. In: Talk presented at a special

workshop on Reconciling ENSO Chronologies for the Past.

3. Li J, Xie S-P, Cook ER, Huang G, D'Arrigo R, Liu F, Ma J, Zheng X-T (2011) Inter-decadal modulation of El Nino amplitude during the past millennium. Nat Clim Change 1:114-118. doi:10.1038/nclimate1086

4. Li J, Xie S-P, Cook ER, Morales MS, Christie DA, Johnson NC, Chen F, D/'Arrigo R et al. (2013) El Nino modulations over the past seven centuries. Nat Clim Change 3:822-826. doi:10.1038/nclimate1936

5. McGregor S, Timmermann A, Timm O (2010) A unified proxy for ENSO and PDO variability since 1650. Climate of the Past 6:1-17. doi:10.5194/cp-6-1-2010

6. Sigl M, Winstrup M, McConnell JR, Welten KC, Plunkett G, Ludlow F, Buntgen U, Caffee M et al. (2015) Timing and climate forcing of volcanic eruptions for the past 2,500 years. Nature 523:543-549. doi:10.1038/nature14565

7. Stahle DW, Cleaveland MK, Therrell MD, Gay DA, D'Arrigo RD, Krusic PJ, Cook ER, Allan RJ et al. (1998) Experimental Dendroclimatic Reconstruction of the Southern Oscillation. B Am Meteorol Soc 79:2137-2152. doi:10.1175/1520-0477(1998)079<2137:edrots>2.0.co;2

8. Wilson R, Cook E, D'Arrigo R, Riedwyl N, Evans MN, Tudhope A, Allan R (2010) Reconstructing ENSO: the influence of method, proxy data, climate forcing and tele-connections. J Quaternary Sci 25:62-78. doi:10.1002/jqs.1297

---

## Author Response (AR2)

**Response to the review comments on "Decadal variation of extreme drought and flood in North China revealed by documentary-based seasonal precipitation reconstruction for the past 300 years" by Jingyun Zheng et al**

Dear editors and reviewers,

Thank you for your valuable comments and thoughtful suggestions on our manuscript. Following your comments on the manuscript, we made careful revisions, and the point-to-point response of the comments is listed below. We hope these revisions would make this manuscript to be more acceptable for publication. Please feel free to contact me if you have any questions.

Many thanks again. With best wishes

Sincerely yours,

Jingyun Zheng

**Anonymous Referee #1**

**General comments In this paper, extreme drought and flood events in North China during 1736-2000 have been identified with seasonal precipitation reconstructions of 17 sites, and then compared with ENSO episodes and large volcanic eruptions. Higher temporary and spatial resolution of precipitation reconstructions allows the authors to identify more extreme events than the previous studies and eliminate some ones that intensity may have been overestimated (as discussed in the section 3.1). This work is helpful for better understanding of the climatic extremes in the past and their driving mechanisms. Before acceptance and publication, some minor revisions are still necessary.**

**Specific comments 1. This paper contains two main parts, identification of extreme drought/flood events, and the relationship between these events and ENSO/volcanic eruptions, however, in the Introduction (Section 1), the authors just introduce the study progress of historical climate change and extreme events. The previous studies cited in the Section 3.3 (from page 6 line 26 to page 7 line 2) which are about the relationship between climatic extremes and ENSO/volcanic eruptions could be moved into the Introduction, and more relevant literature should be added.**

Accepted and revised. The more information about the study on the relationship between climatic extremes in North China Plain and ENSO/volcanic eruptions have been summarized and added. (P3, L10-18)

*Meanwhile, several studies had argued that the anomalous precipitation in NCP, especially the occurrence of drought was related to El Niño and large volcanic eruption. For instance, from the observation data since 1951, it was found that the rainfall decreased over northern China (including NCP and the adjacent area at north and west) not only in the summer and autumn of El Niño developing year (Wu et al, 2003; Lu, 2005; Zhai et al., 2016), but also in the summer when El Niño decayed (Feng et al., 2014; Zhang et al., 2017). From the documentary-based reconstruction of yearly dryness/wetness grade for the past 500 years, Chen and Yang (2013) argued that the occurrence of drought in northern China was in synchronous with El Niño events in the context of decadal variations. Shen et al. (2007) found that the most three exceptional*

*drought events over eastern China occurred in 1586–1589, 1638–1641 and 1965–1966, with 50% or more summer rainfall reduction in the droughty centres, which might be triggered by large volcanic eruptions and amplified by the El Niño events.*

**2. In the Section 2.1, the authors introduce the seasonal precipitation reconstruction which is from their previous work (page 3 line 10-23), and more details of this work, especially the description of the quantitative precipitation records of Yu-Xue-Fen-Cun during the Qing Dynasty and the methods of reconstruction, would be helpful for better evaluation of the scientific significance of this paper.**

Accepted and revised. The description of the quantitative precipitation records of Yu-Xue-Fen-Cun during the Qing Dynasty and the methods of reconstruction on rainfall from Yue-Xue-Fen-Cun records has been added. (P3, L30-P4, L17)

*This dataset was reconstructed from a unique historical archive called Yu-Xue-Fen-Cun, i.e., the quantitative records (in Chinese length units of "Fen" and "Cun"; 1 Fen = 0.32 cm approximately; 1 Cun = 10 Fens) of rainfall (i.e. "Yu") and snowfall (i.e. "Xue") reported in memos to the emperor from local officials in Qing Dynasty (1644–1911), together with available instrumental data. Noting that the rainfall reported in Yue-Xue-Fen-Cun was measured as the infiltration depth (in units of Cun and Fen) from the dry-wet soil boundary layer to the ground surface by digging the soil with a shovel in the flat farmland after each rainfall event. The snowfall was measured as the depth on the surface after each snowfall event, which is similar to the observation of snowfall depth at modern weather station. Besides the quantitative measurements of rainfall and snowfall, the Yu-Xue-Fen-Cun also reported other information, such as the dates and intensity for each precipitation event, the amount of rain or snow days, the summation of infiltration depth or snowfall depth, and the qualitative descriptions within a limited duration (e.g., one month or season) (Ge et al., 2005). For the rainfall reconstruction from historical records, the method was the Green-Ampt infiltration model under surface water balance equation:*

$$P = (\theta s - \theta i) \times \rho \times Zf / \beta$$

*where P is the precipitation; θs, θi, ρ, Zf and β are the soil saturated moisture content, the initial moisture content before rainfall, the apparent specific gravity of the soil, the depth of infiltration (i.e. Yu-Fen-Cun in historical times), and the infiltration rate, respectively, which can be obtained from modern local agrometeorological stations based on the types of soil texture and the seasonality of climate at each site. This is because the physical properties of farmland soil and the seasonality of climate are supposed to remain constant generally over the past 300 years. Moreover, the field infiltration experiment was used for the validation of reconstruction model, which showed that the predicted R2 (i.e., explained variance) for the rainfall reconstruction reached to 87%.*

**3. In the Section 2.2 (Method), the method how the authors identify the extreme drought/flood events should be described more clearly. For example, what does the phrase "these years" (in page 4 line 9-10) mean, all the years in 1736-2000, or just the 10 extreme years in 1951-2000 mentioned in the last sentence? The methods used for comparing the extreme drought/flood events with the ENSO chronology and the large volcanic eruption chronology also need some supplements.**

Accepted and revised. The related sentences have been rewritten (P5, L11-14). The methods used for comparing the extreme drought/flood events with the ENSO chronology and the large volcanic eruption chronology have also been added (P5, L15-17).

*Therefore, we use the minimum percentage of severe and extreme drought (flood) sites among these extreme years in 1951–2000, i.e., 35% (35%) and 29% (24%) of all sites experiencing severe and extreme drought (flood) respectively, as the criteria to identify the regional extreme drought/flood events in 1736–2000 (Fig. 2b).*

*Furthermore, we compare the extreme drought/flood events with the El Niño/La Niña chronology and the large volcanic eruption chronology using the contingency table, to illustrate the characteristics of connections between extreme drought/flood events, and El Niño/La Niña episodes, and large volcanic eruptions, respectively.*

**4. In the Section 3.3, the authors give the probabilities of the occurrences of extreme events with ENSO events and large volcanic eruptions, however, no driving mechanism. Further discussion on how ENSO and volcanic eruption impact the climate change and extreme events might be necessary for this paper. What are the opinions in the previous studies, and why the results of this paper are different?**

Accepted and revised. The discussion on how El Niño impact the climate change and extreme events have been added based on the previous studies (P8, L8-23).

*In addition, many previous studies from observation found that the El Niño could usually cause precipitation evidently decreasing in northern China not only simultaneously, but also in the subsequent summer after an El Niño year (e.g., Wu et al., 2003; Lu, 2005, Zhang et al, 2017). Corresponding to the recent strong El Niño event in 2015-2016, the summer precipitation decreased by 20% to 50% over the northern China (Zhai et al. 2016). Our result confirmed their findings. As suggested by previous studies based on observations and simulations, the mechanism of impact of El Niño on precipitation in NCP can be summarized as follows. In the developing stage of El Niño episode, weakened Walker Circulation could restrain the India summer monsoon and further trigger up an anomalous barotropic cyclone over the East Asian by modulating the wind fields from the western of Tibetan Plateau and affecting the mid-latitude Asian wave pattern along 30°N–50°N. Correspondingly, NCP is located at the area affected by the local downdraft airflow of that anomalous barotropic cyclone. Thus, the rainfall decreases at summer and autumn over NCP (Wu et al, 2003; Lu, 2005). While in the peak and decaying year of El Niño, the warm SST anomaly in tropical Indian Ocean can trigger up Kelvin waves to induce an anomalous easterlies in equatorial atmosphere and the anomalous anticyclone over western North Pacific (Xie et al., 2009). Since the southern flank of western Pacific subtropical high (WPSH) prevail easterly winds, the enhanced easterlies lead to the southward shift of WPSH (Song and Zhou, 2014). The southward shift of WPSH consequently dampens the moisture transport from the Indian summer monsoon to eastern China, resulting in a significant decrease of rainfall in NCP (Zhang et al., 2017).*

**Technical corrections 1. This paper might need to be read and corrected by a native English speaker.**

Accepted. We had done it before our first submission.

WILEY

Wiley Editing Services

**LANGUAGE EDITING**

**CERTIFICATE**

This document certifies that the manuscript listed below was edited for proper English language, grammar, punctuation, spelling, and overall style by one or more of the highly qualified native English speaking editors at Wiley Editing Services.

**Manuscript title:**

Decadal variation of extreme drought and flood in North China revealed by documentary-based seasonal precipitation reconstruction for the past 300 years

**Authors:**

Jingyun Zheng, et al

**Date Issued:**

February 5, 2018

**Certificate Verification Key:**

7901-49EC-24A6-B8F2-54AP

This certificate may be verified at https://secure.wileyeditingservices.com/certificate. This document certifies that the manuscript listed above was edited for proper English language, grammar, punctuation, spelling, and overall style. Neither the research content nor the authors' intentions were altered in any way during the editing process. Documents receiving this certification should be English-ready for publication; however, the author has the ability to accept or reject our suggestions and changes. If you have any questions or concerns about this document or certification, please contact help@wileyeditingservices.com.

[Figure]

Wiley Publishing Services is a service of Wiley Publishing. Wiley's Scientific, Technical, Medical, and Scholarly (STMS) business serves the world's research and scholarly communities, and is the largest publisher for professional and scholarly societies. Wiley is committed to providing high quality services for researchers. To find out more about Wiley Editing Services, visit wileyeditingservices.com. To learn more about our other author services provided by Wiley Publishing, visit authorservices.wiley.com.

**2. In page 4 line 10, the sentence "35% (35%) and 29% (24%) for severe and extreme drought (flood) respectively" should be checked, because according to the Fig.2, 35% (35%) is probably the criteria to identify the extreme drought (flood) events, but not severe ones.**

Accepted and checked. The criteria of "35% (35%) and 29% (24%) for severe and extreme drought (flood) respectively" is right. To be better understood, we have rewritten the sentence (P5, L11-14) and modified figure 2.

*Therefore, we use the minimum percentage of severe and extreme drought (flood) sites among these extreme years in 1951–2000, i.e., 35% (35%) and 29% (24%) of all sites experiencing severe and extreme drought (flood) respectively, as the criteria to identify the regional extreme drought/flood events in 1736–2000 (Fig. 2b).*

[Figure]

Figure 2. Percentage of sites with extreme and severe drought/flood occurred over North China in 1736 – 2000. Dash line: the criteria to identify the regional extreme drought/flood events respectively. ↑: The year of extreme drought/flood events which were not reported in previous studies (Chen and Yang, 2013; Hao et al., 2010a, b; Shen et al., 2007, 2008; Zhang, 2005; Zheng et al., 2006).

**3. For the Fig.3 (page 13), a legend might be necessary to describe the symbols appearing on the figure.**

Accepted. The legend has been added and the modified figure is shown below.

[Figure]

Figure 3. The box-whisker plot of seasonal precipitation anomaly percentage among sites for each extreme drought (a) and flood (b) event.

**Anonymous Referee #2**

Dear editor and authors of the manuscript "Decadal variation of extreme drought and flood in North China revealed by documentary-based seasonal precipitation reconstruction for the past 300 years". To the best of my knowledge, the Yu-Xue-Fen-Cun record is the best proxy data in Eastern China with seasonal resolution and significant relationship to the instrumental precipitation variability. This work chooses the study area at the margin of the Eastern Asian summer monsoon, which will have a critical implication to understand the monsoon variability over the past several hundred years. The idea to definite and identify the extreme drought/flood events during the different centuries is novel in paleoclimate reconstruction community. The results are completely supported by the data analysis. Thus, I suggest that the manuscript should be accepted for publication after a minor revision.

**Main comments:**

**1. The explanation of the features of the extreme drought/flood events in North China could be further inferred. e.g. 1) The higher probability of the extreme drought events mostly followed the El Niño activity, which implied that the El Niño' may triggered the drought event in North China.**

Accepted and revised. The discussion on how El Ni*ñ*o impact the climate change and extreme events have been added based on the previous studies (P8, L8-23).

*In addition, many previous studies from observation found that the El Niño could usually cause precipitation evidently decreasing in northern China not only simultaneously, but also in the subsequent summer after an El Niño year (e.g., Wu et al., 2003; Lu, 2005, Zhang et al, 2017). Corresponding to the recent strong El Niño event in 2015-2016, the summer precipitation decreased by 20% to 50% over the northern China (Zhai et al. 2016). Our result confirmed their findings. As suggested by previous studies based on observations and simulations, the mechanism of impact of El Niño on precipitation in NCP can be summarized as follows. In the developing stage of El Niño episode, weakened Walker Circulation could restrain the India summer monsoon and further trigger up an anomalous barotropic cyclone over the East Asian by modulating the wind fields from the western of Tibetan Plateau and affecting the mid-latitude Asian wave pattern along 30°N–50°N. Correspondingly, NCP is located at the area affected by the local downdraft airflow of that anomalous barotropic cyclone. Thus, the rainfall decreases at summer and autumn over NCP (Wu et al, 2003; Lu, 2005). While in the peak and decaying year of El Niño, the warm SST anomaly in tropical Indian Ocean can trigger up Kelvin waves to induce an anomalous easterlies in equatorial atmosphere and the anomalous anticyclone over western North Pacific (Xie et al., 2009). Since the southern flank of western Pacific subtropical high (WPSH) prevail easterly winds, the enhanced easterlies lead to the southward shift of WPSH (Song and Zhou, 2014). The southward shift of WPSH consequently dampens the moisture transport from the Indian summer monsoon to eastern China, resulting in a significant decrease of rainfall in NCP (Zhang et al., 2017).*

**2) The precipitation variability in summer and autumn is stronger than other seasons, the reason may be that the precipitation during the flood season accounts for a large proportion of annual precipitation.**

Accepted and revised. (P2, L19-20)

*This region is located at the margin of the Eastern Asian summer monsoon (EASM) and has a sub-humid warm temperate climate with the summer and autumn precipitation accounts for 80% annual precipitation approximately.*

**3) Some events have not mentioned in the previous studies, which indicates that the Yu-Xue-Fen-Cun has more accurate than other proxy record, e.g. the drought/flood grads, or the definition of the drought/flood events in this study is different from the other studies.**

Accepted. We have added it in the revised version. (P6, L14-16)

*This is mainly because the series of seasonal precipitation were reconstructed from Yu-Xue-Fen-Cun, thus more accurate in seasons and sub-regions than the series of yearly dryness/wetness grade used in other studies.*

**2. The phrase 'decadal variation' usually means that the record is smoothed by the 7 or 9 years low-pass filter to show the decadal variability. Here, the inter-annual events are analyzed during the different centuries. Thus, the phrase 'decadal variation' may not be a best choice.**

Accepted and revised. The "decadal variation" has been replaced by "variation".

**3. The reason for selecting the study area should be further emphasized in climate research, e.g. the North China is located at the margin of the Eastern Asian summer monsoon, which is more sensitive to the global climate change.**

Accepted and revised. The reason for selecting the study area have been added. (P2, L19-23)

*This region is located at the margin of the Eastern Asian summer monsoon (EASM) and has a sub-humid warm temperate climate with the summer and autumn precipitation accounts for 80% annual precipitation approximately. As revealed by other studies (e.g., Wang, 2002; Wang and He, 2015; Ding and Wang, 2016), the climate in this region is sensitive to the global change and the rainfall decreased dramatically from the late 1970s, which had caused the evident impacts on water resources and agriculture when the EASM became weakened.*

**Specific Comments:**
**1. Page 1, The seasonal features of the drought/flood events should be added in the abstract to emphasize the advantage of the Yu-Xue-Fen-Cun record.**

Accepted and revised. The seasonal features of the drought/flood events have been added. (P1, L14-18)

*For most of these extreme drought (flood) events, precipitation decreased (increased) evidently at most of sites for four seasons, especially for summer and autumn. But in drought years of 1902 and 1981, precipitation only decreased in summer slightly, while it decreased evidently in other three seasons. Similarly, the precipitation anomalies for different seasons at different sites also*

*existed in several extreme flood years, such as 1794, 1823, 1867, 1872 and 1961.*

**2. Page 2, line 14. Is the 'larger precipitation anomaly percentage' more suitable? There as on is that the precipitation in south China is larger than one in North China.**

Accepted. We use "large precipitation variability", not "large precipitation" in the manuscript. (P2, L18)

*…especially in regions with large precipitation variability and dense population, such as North China Plain (NCP).*

**3. Page 2, lines 16-35. The definition of drought/flood events should be added since the different definition will obtain the different classifications. e.g. Zhang (2005) points out that there are only 15 events in North China over the past 1000 years in Page 2, lines 18-19, but there are five events during some centuries in this study.**

Accepted. We have added more information in order to make these definitions clear. (P2, L25-P3, L1)

*…Zhang (2005) identified the 15 severe persistent ($\geq$ 3 years) drought events that occurred in NCP and surrounding area over the past 1000 years, and found that the most of historical persistent drought events (i.e., those before 1950) were more severe than that occurred in 1951–2000. Shen et al. (2008) investigated the characteristics of anomalous precipitation events during the past five centuries over eastern China (including North China and the mid-lower Yangtze River Valley), using the dataset of yearly dryness/wetness grade over eastern China (CAMS, 1981), and found that in NCP, the high frequency of severe and extreme (5% and 2.5% occurrence probabilities in 1470–2000 respectively, same as for droughts) floods occurred in the 1650s, 1660s, 1750s, 1760s, 1820s and 1890s, whereas the period of 1580–1650 and the 1990s witnessed more severe and extreme droughts. Our previous studies (Hao et al., 2010a; Zheng et al., 2006) identified the extreme drought/flood events (10% occurrence probabilities in 1951–2000) and extreme persistent drought/flood events ($\geq$ 3 years, 5% occurrence probabilities over all 2000 years) for the past 2000 years by merging the yearly dryness/wetness grade from individual sites.*

**4. Page 3, lines 1-3. The Yu-Xue-Fen-Cun record is not firstly used in this study, thus, this sentence is inappropriate.**

Accepted. The sentence has been rewritten. (P3, L19-20)

*However, most of these studies were performed from the yearly dryness/wetness grade data and relevant historical descriptions or the limited instrumental period.*

**5. Page 3, line 13. Here, the longitude should be a range from 108°E to another longitude (120°E?).**

Accepted and revised. (P3, L28)

*…located in North China Plain (34°N–39°N, 108°E–120°E approximately)…*

**6. Page 3, lines 23-24. Please explain the reason for selecting Gergis and Fowler's ENSO index, because there are some other ENSO index reconstructions in the past millennium (Braganza et al. 2009; Cook et al. 2008; Li et al. 2011; Li et al. 2013; McGregor et al. 2010; Stahle et al. 1998; Wilson et al. 2010).**

Accepted and revised. The reason is added. (P4, L24-28)

*There are some other ENSO index reconstructions in the past millennium (e.g., Stahle et al. 1998; Braganza et al. 2009; McGregor et al. 2010; Wilson et al. 2010; Li et al. 2011, 2013). The reason for our study using the reconstruction by Gergis and Fowler (2009) is that their result was compiled as a chronology for each El Niño and La Niña episode rather than the ENSO index; thus, it is more appropriate for comparison with the extreme drought/flood event by event.*

**7. Page 3, lines 27-28. Regarding to the volcanic eruption index, a new reconstruction is encouraged to use (Sigl et al. 2015). Moreover, it would be interesting that the other degrees of eruptions are assessed, or the Southern Hemisphere eruptions are excluded.**

Accepted and revised. The reason for selecting the chronology of large volcanic eruption used in our study has been added (P4, L33-P5, L1)

*Compared to the other reconstruction on volcanic eruption index (e.g., Sigl et al., 2015), this chronology contains each volcanic eruption event, which is convenient to compare with the extreme drought/flood event.*

**8. Page 5, line 11. There is an extra period '.'.**

Accepted and revised.

**9. Please change 'El Nino' with 'El Niño'.**

Accepted and revised.

**10. If the Yu-Xue-Fen-Cun record is archived in a published repository, which will be a huge contribution for the paleoclimate integration community.**

Accepted. We agree too.

**Comments by Editor, Marit-Solveig Seidenkrantz**

**Your manuscript has now been seen by two reviewers. Both reviewers are overall positive about your work, recommending publication after some moderate corrections, and there is no doubt that a better understanding on large-scale drought events is of significant societal interest. I therefore invite you to resubmit your manuscript after moderate revision, taking all comments by the reviewers into account.**

All comments by the reviewers have been accepted for the revision with point-to-point reply above.

**In addition to addressing the reviewers' comments, I would ask you to expand further on your explanations to the methodologies, as many of the journal readers (myself included) are less familiar with the methodologies used in your study. I would in particular ask you to explain more clearly your methods of identifying droughts (methods section 2.2), show more clearly how you obtain your results in (data, 3.1) (here a figure would be welcome), and explain the statistical methods used when comparing drought events and climate phenomena such as ENSO. Although you are already well on your way according to your comments to the reviewers' comments, I would still ask you to expand on this part. Also, I would recommend you to illustrate the comparison through e.g. cross-plots or some other figure, which would make the data visually easier to understand and evaluate for the reader.**

Accepted and revised. The methods of identifying extreme droughts/flood have been explained more clearly with the cross-plots in Fig. 2. The statistical methods are expanded as follows: (P5, L15-23)

*Furthermore, we compare the extreme drought/flood events with the El Niño/La Niña chronology and the large volcanic eruption chronology using the contingency table, to illustrate the characteristics of connections between extreme drought/flood events, and El Niño/La Niña episodes, and large volcanic eruptions, respectively. For example, to detect whether the frequency of extreme drought becomes higher in the years after El Niño events, we create the contingency table by calculating the numbers of occurrences with extreme drought and El Niño in the previous year, extreme drought and no El Niño in the previous year, no extreme drought but El Niño in the previous year, no extreme drought and El Niño in the previous year. Then, the Chi-square test (χ2) is adopted to test the significance. Similarly, the contingency table and test are also performed for other cases with the occurrences of extreme drought or flood events in NCP associated with the events of ENSO or large volcanic eruption, respectively.*

**I could also not help wondering if you can identify any long-term pattern in your data? Based on your manuscript, if does not seem so, but it would be beneficial if you could discuss this possibility.**

Accepted and discussed. The discussion on the possibility from other historical documents to identify the long-term pattern on regional extreme climate events and their association with

anomalous forcings were added as follows. (P9, L16-24)

*The archive of Yu-Xue-Fen-Cun provided the quantitative records on rainfall and snowfall with high spatial and temporal resolution for seasonal precipitation reconstruction from 1736, which enabled us to investigate the variation of extreme drought and flood with seasonal features. Besides Yu-Xue-Fen-Cun, China also has other historical documents (e.g., local gazettes, official histories, weather diaries, etc.) with abundant and well-dated records on weather, anomalous climate, climate-related natural disasters, the impacts of weather and climate anomalies, as well as phenology, which have been used for the reconstruction of past climates extended to more than thousands of years at resolution of annual to decadal scale (Ge et al., 2016). Thus, most of them could further be applied to identify the regional extreme climate events at yearly resolution or prominent decadal climate anomalies before 1736, and to investigate long-term pattern on the occurrences of regional extreme climate events associated with anomalous forcings in future works.*

**As for your answer to point 4 to Reviewer 1's comments, please remember that you can only assume and suggest a driving mechanism (which I agree with the reviewer was lacking), but you cannot be sure, that your suggestion is correct. So consider your wording. Also, comparison to modern weather data may here be useful, if possible.**

Accepted and revised. We use "As suggested by previous studies based on observations and simulations" to present the driving mechanism. The comparison to modern weather data was added as follows: (P8, L8-23)

*In addition, many previous studies from observation found that the El Niño could usually cause precipitation evidently decreasing in northern China not only simultaneously, but also in the subsequent summer after an El Niño year (e.g., Wu et al., 2003; Lu, 2005, Zhang et al, 2017). Corresponding to the recent strong El Niño event in 2015-2016, the summer precipitation decreased by 20% to 50% over the northern China (Zhai et al. 2016). Our result confirmed their findings. As suggested by previous studies based on observations and simulations, the mechanism of impact of El Niño on precipitation in NCP can be summarized as follows. In the developing stage of El Niño episode, weakened Walker Circulation could restrain the India summer monsoon and further trigger up an anomalous barotropic cyclone over the East Asian by modulating the wind fields from the western of Tibetan Plateau and affecting the mid-latitude Asian wave pattern along 30°N–50°N. Correspondingly, NCP is located at the area affected by the local downdraft airflow of that anomalous barotropic cyclone. Thus, the rainfall decreases at summer and autumn over NCP (Wu et al, 2003; Lu, 2005). While in the peak and decaying year of El Niño, the warm SST anomaly in tropical Indian Ocean can trigger up Kelvin waves to induce an anomalous easterlies in equatorial atmosphere and the anomalous anticyclone over western North Pacific (Xie et al., 2009). Since the southern flank of western Pacific subtropical high (WPSH) prevail easterly winds, the enhanced easterlies lead to the southward shift of WPSH (Song and Zhou, 2014). The southward shift of WPSH consequently dampens the moisture transport from the Indian summer monsoon to eastern China, resulting in a significant decrease of rainfall in NCP (Zhang et al., 2017).*

**Please mark in the text, any changes that you make to the manuscript.**

*All revisions are marked in red.*

**Comments by Editor, Marit-Solveig Seidenkrantz**

**In your answer (conclusion), you just mention the possibility of using the data for studies of longer-term patterns. But what I really hoped you would do, was to check if there is indeed any indication in your data on whether the frequency of drought/floods has changed over time? As your data reach back in time prior to the start of industrialization, information on potential changes (or not) in droughts/flood frequency/severety would be very valuable, if possible.**

Accepted. We examined the long-term trend on frequencies of extreme drought, flood, and sum of drought and flood per decade from 1740s to 1990s. The results showed only the frequency of extreme drought showed a slightly increase trend (0.29 times per 100-yr) from 1740s to 1990s, but it was not statistically significant. This statement was added in P7, L8-9. The frequency of extreme flood, and sum of drought and flood didn't show any trend. Thus we did not make any statement on them.

References added in the revised version:

[revised manuscript text omitted]